# Increased CD47 and MHC Class I Inhibitory Signals Expression in Senescent CD1 Primary Mouse Lung Fibroblasts

**DOI:** 10.3390/ijms221910215

**Published:** 2021-09-23

**Authors:** Elisa Hernández-Mercado, Jessica Lakshmi Prieto-Chávez, Lourdes Andrea Arriaga-Pizano, Salomon Hernández-Gutierrez, Fela Mendlovic, Mina Königsberg, Norma Edith López-Díazguerrero

**Affiliations:** 1Laboratorio de Bioenergetica y Envejecimiento Celular, Departamento de Ciencias de la Salud, Universidad Autónoma Metropolitana-Iztapalapa, Mexico City 09340, Mexico; eli2be3@hotmail.com (E.H.-M.); mkf@xanum.uam.mx (M.K.); 2Posgrado en Biología Experimental, Universidad Autónoma Metropolitana-Iztapalapa, Mexico City 09340, Mexico; 3Hospital de Especialidades CMN “Siglo XXI”, IMSS, Mexico City 06720, Mexico; lakshmi.litmus@hotmail.com (J.L.P.-C.); landapi@hotmail.com (L.A.A.-P.); 4Laboratorio de Biología Molecular, Escuela de Medicina, Universidad Panamericana, Mexico City 04510, Mexico; shernand@up.edu.mx; 5Facultad de Medicina, UNAM, Mexico City 04360, Mexico; fmendlo@yahoo.com; 6Facultad de Ciencias de la Salud, Universidad Anáhuac Mexico Norte, Mexico City 04510, Mexico

**Keywords:** senescent cell clearance regulation, CD47, MHC class I, calreticulin

## Abstract

Cellular senescence is more than a proliferative arrest in response to various stimuli. Senescent cells (SC) participate in several physiological processes, and their adequate removal is essential to maintain tissue and organism homeostasis. However, SC accumulation in aging and age-related diseases alters the tissue microenvironment leading to deterioration. The immune system clears the SC, but the specific scenarios and mechanisms related to recognizing and eliminating them are unknown. Hence, we aimed to evaluate the existence of three regulatory signals of phagocytic function, CD47, major histocompatibility complex class I (MHC-I), and calreticulin, present in the membrane of SC. Therefore, primary fibroblasts were isolated from CD1 female mice lungs, and stress-induced premature senescence (SIPS) was induced with hydrogen peroxide. Replicative senescence (RS) was used as a second senescent model. Our results revealed a considerable increment of CD47 and MHC-I in RS and SIPS fibroblasts. At the same time, no significant changes were found in calreticulin, suggesting that those signals might be associated with evading immune system recognition and thus averting senescent cells clearance.

## 1. Introduction

Cellular senescence is a state of irreversible proliferation arrest attained due to physiological or stress stimuli [1]. There are multiple inducing stimuli, as well as various types of senescence: replicative senescence (RS) is generated mainly by telomere shortening and genomic instability; physiological senescence is observed during the processes of regeneration, wound-healing, and embryonic development; stress-induced premature senescence (SIPS) is generated by multiple stress factors such as reactive oxygen species (ROS), drugs, DNA damage; Oncogene-induced senescence (OIS) occurs by activation of oncogenes such as RAS, RAF, and BRAF; senescence induced by ionizing radiation (IR) where DNA damage is due to double-strand break [2]. Senescent cells (SC) are characterized by phenotype alterations such as increased cell size, elongated and flattened appearance, high activity of senescence-associated β-galactosidase (SA-β-gal) at pH 6, increased granularity or cellular complexity, metabolic aberrations, chromatin remodeling, altered gene expression and regulation, changes in mitochondrial dynamics, decreased proteostasis and autophagy [3].

The SC are metabolically active but do not respond to mitogenic or apoptotic stimuli, which could explain, in some cases, their accumulation in tissues and organs. In particular, they possess a secretome known as the senescence-associated secretory phenotype (SASP) with complex production of molecules such as cytokines, chemokines, metalloproteinases, and growth factors, among others [4], which has beneficial and detrimental effects on the tissues [5]. 

Several studies have linked the accumulation of SC with the appearance and progression of multiple chronic-degenerative diseases associated with aging, such as cancer, Alzheimer’s, Parkinson’s, arthritis, osteoarthritis, diabetes, cataracts, and benign prostatic hyperplasia [6,7,8,9]. Due to the involvement of SC in various diseases, their identification and elimination are essential [10].

The retinoblastoma (RB) family and p53 participate in the cell cycle arrest in SC. The accumulation of CDK2 inhibitor p21WAF1/Cip1 (CDKN1A) and CDK4/6 inhibitor p16INK4A (CDKN2A) results in the activation of the RB family proteins, the inhibition of E2F, and finally, the irreversible arrest of the cell cycle [11,12]. Apart from the expression of p16 and p21, other persistent changes in SC are used as molecular markers in vitro and in vivo experiments, such as p27, the Bcl-2 family proteins (Bcl-2 and Bcl-XL) with antiapoptotic activity, DNA damage markers such as phosphorylated histone H2AX, decreased nuclear envelope protein Lamin B1, and SASP expression [13]. Several markers are used to identify SC because there is currently no universal marker available.

SC are cleared by the immune system at the end of their physiological functions, although the exact mechanisms are not fully understood. However, SC tend to accumulate in the tissue during aging and other pathologies, altering its microenvironment and contributing to cellular deterioration [1,4,5,14]. SC accumulation may result from an aged immune system [15]; or due to the loss of superficial ligands preventing the immune cells from clearing them [16]. SC removal is partially attributed to several immunomodulators within the SASP, such as monocyte chemoattractant protein-1 (MCP-1), macrophage inflammatory protein-1 alpha (MIP-1α), chemokine CCL5 (RANTES), and other molecules [4,17]. SC also express specific signals on their cell surface that regulate the immune recognition by macrophages and NK cells [18,19]. However, the specific molecules and mechanisms are unclear, considering the heterogeneity in SC conditions [16]. 

It has been described in murine cells that the CD47 and MHC-I markers mediate the “do not eat-me” antiphagocytic signals in cancer cells [20,21], so it might be possible that these same molecules were increased on SC membranes, blocking the regulatory signals for macrophages thus evading the immune response. Moreover, the CD47 loss in aged erythrocytes has been related to increased phagocytosis [22]. Likewise, MHC-I has been considered an immunotherapeutic target against cancer [21] because it protects cancer cells from phagocytosis. This protection is mediated by the inhibitory receptor Leukocyte Immunoglobulin-Like Receptor B1 (LILRB1) expressed in macrophages. On the other hand, calreticulin, as an “eat-me” signal in apoptotic cells, functions as a chaperone in the endoplasmic reticulum (ER), but its location on the cell surface can initiate cellular immune clearance [23]. 

Therefore, we aimed to evaluate these positive and negative phagocytic signals in SC. Several in vitro models have been used to study cellular senescence, such as RS, which mainly results from telomere shortening in each cell duplication; SIPS attained due to oxidative stress or other insults [24], and oncogenic induced senescence (OIS) [25]. Here, we used RS and SIPS lung CD1 female mice primary fibroblasts to identify CD47, MHC-I, and calreticulin.

Our results showed an increase in the expression of CD47 and MHC-I; however, we did not find a significant change in calreticulin in the senescent models, suggesting that at least these two “do not eat-me” signals might be involved in evading mechanisms that avert SC clearance by the immune system.

## 2. Results

### 2.1. Establishment and Identification of Senescence in RS and SIPS Cell Cultures

The RS and SIPS senescent fibroblast models have already been reported on several occasions by our working group [24,26,27], where more than 60% of cultured cells attained RS on day 21 and SIPS with almost 80% from day 16 to 21 of cultivation. Even so, to confirm the senescent state, SA-β-Gal and p21 increase were evaluated as shown in Figure 1 and Figure 2, corroborating the senescent phenotype. Non-senescent cells (NS) at day nine were used as control.

### 2.2. Senescent Fibroblasts Identification

Side scatter (SSC) and forward scatter (FSC) were determined, and the aggregates and debris were excluded from selecting fibroblasts cell population (Figure 3A). Then, CD45 and CD140a were used as markers to identify fibroblasts [20]. As expected, CD45 positive cells were less than 10%, while almost 100% of cells from the two models were positive for CD140a (Figure 3B). No significant differences were observed among the cell cultures. CD45- and CD140a+ were considered fibroblasts. The FSC provides information on the relative size of the cells, whereas the SSC is analyzed as cell complexity, also called granularity, an indicator for the internal structure of the cell (e.g., nuclear morphology, quantities of organelles) [28,29].

Subsequently, we distinguished the senescence from the proliferating fibroblast populations. Senescent fibroblasts, RS and SIPS, were selected based on their complexity and the brightness due to the C12FDG cleavage by the ß-Gal enzyme. The FSC/SSC dot plot showed the senescent fibroblast population’s displacement based on their complexity and fluorescence intensity associated with SA-ß-Gal (brightness) (Figure 4A). The NS proliferating fibroblasts were less complex and had less SA-ß-Gal, while the RS and SIPS fibroblast populations were significantly more complex and showed higher SA-ß-Gal activity (Figure 4B). 

### 2.3. Calreticulin, CD47, and MHC-I Expression in RS and SIPS Fibroblasts

Figure 5A shows calreticulin expression on the cell surface. No differences were found among NS, RS, and SIPS fibroblasts. Interestingly, CD47 significantly increased in SIPS fibroblasts (* *p* <0.05), and a marginal change in RS fibroblasts was determined (*p* = 0.06) in comparison with NS fibroblasts (Figure 5B). MHC-I was also significantly augmented in RS and SIPS (* *p* <0.05) compared to NS fibroblasts (Figure 5C).

## 3. Discussion

Because SC accumulate in tissues during aging in related pathologies, our group was interested in evaluating several cell surface molecules that, if altered, might be avoiding their clearance. Numerous studies in murine models have suggested that macrophages carry out SC clearance. One of the first reports in this regard showed that SC’s presence in hepatocarcinoma triggers a macrophage-mediated immune response against tumor cells [30]. In 2011, immune cells, including monocytes/macrophages in senescent pre-malignant hepatocytes, were reported [31]. Although the evidence supports that SC induce an immune response to be eliminated, their accumulation has been related to tissue damage and pathologies such as cancer [32]. Therefore, the importance of SC surveillance and clearance is critical due to the possibility of modifying the microenvironment, maintaining tissue integrity, and delaying the appearance of age-related pathologies [33]. Although recent publications report some signals that regulate the immune response against SC, much remains to be determined, such as which signals regulate macrophage-mediated phagocytosis. Macrophages identify and recognize their target through phagocytosis activation or inhibition signals (“eat-me” or “do not-eat-me”, respectively) that are exposed on the cell surface of target cells. In this way, our research focused on identifying these regulatory signals’ presence and levels of surface membrane expression in senescent fibroblast (RS and SIPS) as a possible therapeutic target to clear SC.

One of the most relevant aspects of cellular senescence study is that SC accumulation throughout life contributes to age-associated pathologies [34]. Eliminating SC improves tissues health and increases life expectancy; therefore, studies related to the mechanisms that regulate their elimination are an essential line of research [33]. Several studies have suggested that phagocytes, such as macrophages, carry out SC elimination [30,31]; therefore, it is crucial to understand the signals regulating macrophage-mediated phagocytic function (Figure 5). Hence, our study focused on identifying the presence of three regulatory signals in the SC surface: the “eat-me” calreticulin and the “do not eat-me” CD47 and MHC-I signals in RS and SIPS (Figure 4).

Fibroblast primary cultures have been widely used as in vitro models to study aging. To assure that our cells were indeed fibroblasts, only the CD45- CD140a+ population was used (Figure 3). Subsequently, we identified the senescent fibroblasts by determining the increase in SA-ß-Gal and cellular complexity, as previously described [35,36].

Calreticulin has important implications in health and disease; its functions depend significantly on cell localization in the ER, the cell surface, the extracellular space, and the cytosol [37]. Calreticulin is sensitive to changes in ER stress, DNA damage, and oxidative stress. Calreticulin is present on the surface of healthy cells and does not participate in its uptake by phagocytes because it is usually sensed as a “do not eat me” signal such as CD47. In stressed or dying cells exposed to various extracellular or intracellular stress factors such as hypoxia, high temperature, Ca2+ or pH imbalance, cytokines, or death inducers, calreticulin can appear on the surface of the membrane or in the extracellular space, acquiring characteristics to be eliminated throughout the immune mechanism [38]. Given this scenario, and since no studies have demonstrated the expression of calreticulin on the cell surface of SC, we decided to explore its presence in our study and compare it with non-senescent cells. However, although an expression was found, which we probably could consider basal, we did not found apparent differences between NS, RS, and SIPS (Figure 5A).

There are no studies on membrane surface calreticulin of SC, only as part of SASP. In a study to determine the composition of SASP, in which radiation-induced secretion of senescent lung fibroblasts was evaluated, the authors found high secretion of damage-associated molecular patterns (DAMPs) such as HMGB1 and calreticulin, which are danger signals to initiate inflammatory responses and immune clearance at sites of stress and tissue damage [39]. Calreticulin expression in aged models has been previously described, but contradictory effects have been reported. Erickson et al. studied male rat livers [40] but did not find a decrease in calreticulin with age. In contrast, Boraldi and colleagues [41] found a reduction in calreticulin expression in a RS model of human primary dermal fibroblasts and increased ROS levels. They suggested a decrease in calreticulin and other ER proteins related to the age-dependent accumulation of misfolded proteins and oxidative stress. Studies carried out with cancer cells showed that their phagocytosis might depend on the levels of calreticulin expressed in their cell membrane as well as its secretion [42]. Since calreticulin does not have a transmembrane domain, it is anchored to the membrane by glucan molecules from membrane glycoproteins [43]. Therefore, these molecules might be altered during senescence, preventing their union with calreticulin to the cell membrane, or the MMP secreted by the SASP might be cleaving these unions, as reported for other markers such as NKG2D-ligands [18]. Due to these alterations, it could be thought that “eat-me” signals, such as calreticulin, may be altered in SC associated with pathological processes during aging such as osteoarthritis, Alzheimer’s and Parkinson’s diseases, and cancer [44,45]. Although calreticulin acts as a stimulatory signal to clear cancer cells, it apparently does not increase in SC and primary cultures of non-transformed human cells [46]. Therefore, it will be essential in the future to evaluate whether the secretion of calreticulin in our senescent models could be one pathway for the elimination of SC.

CD47 is a ubiquitously expressed transmembrane protein that can interact with the macrophage receptor SIRPα blocking its phagocytic function by inhibiting the cytoskeleton rearrangement [22,47]. Here, we found an increase in CD47 expression in the SIPS fibroblasts and a marginal increase in the SR fibroblasts compared to NS cells (Figure 5B). No significant differences were found in CD47 expression between RS and SIPS models. CD47 gene is regulated by the transcription factors MYC, NF-κB, and PPAR [48], so NF-κB, which is increased in SC due to the activation of different signaling pathways in response to stress stimuli, might be related to CD47 increment. It is important to emphasize that NF-κB has been widely described in SC as a regulatory protein of the SASP, and NF-κB pathway hyperactivation is one of the transcriptional signatures of aging [49], so this transcription factor might upregulate the expression of CD47 in our senescent models. Besides, CD47 could have diverse functions during cellular senescence. Studies using microvascular endothelial primary cultures from mice brains showed that the signaling of the protein thrombospondin 1 (TSP1)-CD47 inhibited their proliferation and induced RS. Moreover, CD47 deficiency revived the cell cycle and delayed RS [50]. Other studies carried out in breast cancer (MCF7) and colon cancer (LS174T) cell lines induced to senescence by chemotherapy (CIS) showed that p21 can regulate the expression of CD47 and that its union with TSP1 works in order to maintain the senescent state; these kinds of cells escaped senescence by decreasing p21 and CD47 [51].

Increased CD47 has also been reported in cancer cells as part of an immune clearance avoidance mechanism, specifically for macrophages [48], and an antitumor therapy has been proposed by blocking CD47. Thus, CD47 inhibition with an anti-CD47 antibody might increase SC phagocytosis in old tissues and help alleviate age-related conditions. Since CD47 could be involved in maintaining senescence, it would be interesting to assess whether its presence favors SC accumulation in aging.

Cellular homeostasis can be regulated through prophagocytic and antiphagocytic signals such as calreticulin and CD47, respectively. Calreticulin is the dominant signal in multiple human cancers counteracted by CD47 [46]; moreover, in cancer cells that express calreticulin on their cell surface, increased CD47 has been shown to protect them from calreticulin-mediated phagocytosis, and when CD47 is blocked, cancer cells are killed by macrophages [20]. Based on the above, it is understood that the expression of phagocytosis stimulating signals, as calreticulin, might be neutralized by antiphagocytic signals such as CD47. 

Because other stimulating signals of the phagocytic function appear to exist, their counterparts beyond CD47 may also exist. It is known that inhibiting CD47 and SIRPα interaction is insufficient to induce a full increase in macrophage-mediated phagocytosis so that other inhibitory signals might be participating. MHC-I has been proposed as an essential molecule that mediates macrophage-mediated phagocytosis [21]. 

Our results showed significant changes in MHC-I expression in RS and SIPS fibroblasts compared to NS fibroblasts. Like CD47, MHC-I expressed in SC could bind to LILRB1 on the macrophages’ surface, consequently inhibiting phagocyte cytoskeleton rearrangement. However, our results contrast with Pereira et al. [52], who did not find a significant increase in MHC-I expression in X-ray-induced senescent primary human dermal fibroblasts. Nevertheless, the stimulus that induces the senescent state can modify diverse expression patterns, explaining senescent phenotype heterogeneity [24,53].

MHC-I function in senescent cells cannot be reduced to a phagocytic role. This molecule could be participating in the exposure of stress-related antigens for their recognition and elimination by the immune system, since MHC-I is an intracellular stress indicator that notifies leukocytes to present new antigens. The DNA damage and senescent induction pathway may also be related to MHC-I expression, since p53 activates the transcription of endoplasmic reticulum aminopeptidase 1 (ERAP1) mRNA, which is related to increased surface expression of MHC class I [54]. With all this information, we could assume that in aging and related pathologies, the lack of maintenance of the inhibitory signals of phagocytic function and the altered signaling probably cause the senescent cells’ accumulation in tissues generating dysfunction.

Dysregulation between the expression of activation or inhibition signals for clearance of SC by macrophages should be further explored. From our results in SR and SIPS, it is interesting to assume that the increase in MHC-I and CD47 could be related to the evasion of SC from the immune system. However, it is imperative to perform macrophage functional assays with antibodies against CD47 and MHC-I to determine if they are part of the surveillance of SC by the immune system (Figure 6).

Other signals stimulating phagocytosis could be involved in SC clearance; for this reason, it is essential to evaluate other signals related to senescence and the regulation of the immune response of macrophages. For example, in a study by Frescas et al., the IgM isotype antibodies were recovered from BALB/c mice immunized with senescent mouse lung fibroblasts, where clone 94H was able to bind to the modified form of vimentin, which undergoes a post-translational change in cysteine 328 during senescence (C328) forming a malondialdehyde oxidative adduct (MDA). Possibly the MDA-vimentin can be recognized by IgM’s to initiate macrophage-mediated phagocytosis [55]. On the other hand, another possible candidate to evaluate could be the CD36 scavenger receptor described as being overexpressed in epithelial cells and senescent fibroblasts in response to replicative, oncogenic, and chemical stimuli [56]. Apoptotic cells that express CD36 on their surface are eliminated by phagocytosis [57,58], so we suggest that the expression of CD36 in SC could stimulate recognition by macrophages and be phagocytosed. Functional assays should be performed between SC and macrophages to evaluate these and other possible signals and their response.

From what has been discussed above, we concluded that there must exist a careful balance between pro and antiphagocytic signals to eliminate SC by the immune system, and our results show that CD47 and MHC class I are part of this complex process, despite having other functions. Therefore, if these inhibitory signals were blocked to eliminate SC, the consequences would have to be evaluated in the context of aging and related pathologies.

## 4. Materials and Methods

### 4.1. Animals

CD1 female mice (Mus musculus) were obtained from the closed breeding colony at the Universidad Autonoma Metropolitana-Iztapalapa. 

### 4.2. Fibroblasts Primary Culture

Primary lung fibroblasts were isolated from newborn CD1 female mice (3–5 days) as previously described by our group [26,27].

### 4.3. Replicative Senescence and Stress-Induced Premature Senescence

Fibroblasts attained in vitro RS on day 21 since they displayed all senescent markers, as previously reported by our group [24,26,27,59]. As described before [26,60], SIPS was induced by incubating the fibroblasts for 2 h with 75 μM H_2_O_2_ (H_2_O_2_ 30 % (*w*/*w*) Sigma-Aldrich, St. Louis, MO, USA). Eighty percent of the treated fibroblasts attained SIPS on day 16. Non-Senescent cells (NS) were used as controls on day 9.

### 4.4. Senescence Markers

The senescence-associated β-galactosidase activity (SA-β-Gal) and p21 increment were determined to confirm the senescent phenotype at days 9, 12, 16, and 21 as previously described [61,62].

SA-β-Gal assay: the gradual elevation overtime activity of β-Gal in SC was determined as described by Dimri and colleagues in four-time points at days 9, 12, 16, and 21 [61]. Cells were fixed with 4% paraformaldehyde for 15 min and washed three times with phosphate-buffered saline (PBS Sigma-Aldrich (Sigma-Aldrich St. Louis, MO, USA) 1X. Subsequently, three wells were stained from each specific time point, with 300 µL of X-gal solution pH6 was added. The cells were incubated at 37 °C overnight. The number of SA-β-Gal positive cells was determined by counting at least 100 cells per well under an inverted phase-contrast microscope. The number of cells positive for SA-β-Gal activity is reported as percentages of the total number of cells scored.

Detection of p21 by immunofluorescence. Seven days after isolation, primary lung fibroblasts were trypsinized and seeded on coverslips at 30 × 10^3^ cells/coverslip cell density. After 48 h, cells were treated to induce SIPS or untreated in the case of RS. Cells at days 9, 12, 16, and 21 of cell culture of the SIPS and SR models were fixed with 4% formaldehyde. Cells were incubated with 200 mL of protein blocker for 10 min, washed with PBS 1X, and incubated for 1 h with anti-p21 dilution 1:100 (Clone: F-5, Santa Cruz Biotechnology Inc., Santa Cruz, CA, USA). Cells were washed twice with PBS 1X-tween; subsequently, they were incubated with the secondary antibody Alexa Fluor 594 (Invitrogen) for an hour, and then they were washed. Each coverslip with the cells was collected and mounted on a slide where a 20 µL mounting solution (Dako Cytomation, Glostrup, Denmark) and 0.01% DAPI (4,6-diamino-2-phenylindole, Life Technologies, Eugene, OR, USA) was added. The preparations were observed in a confocal microscope LSM 510 META Axioplan 2 confocal laser scanning microscope (Carl Zeiss; Jena, Germany).

### 4.5. Primary Fibroblasts Identification by Flow Cytometry

Due to the heterogeneous nature of primary cultures, it was necessary to use two markers to identify the fibroblasts, so flow cytometry experiments were performed on a BD Influx Cell Sorter-BD Facs Software (St. José, CA, USA). Data were analyzed in FlowJo software version 10.4.2 (TreeStar, Ashland, OR, USA). At least 10 × 10^4^ events were acquired and analyzed (Fluorescence Minus One control). The cellular models reported above NS, RS, and SIPS were acquired on the same day for their comparison. 0.25 µg/million cells were incubated with anti-CD45, a marker for hematopoietic cells (Biolegend, San Diego, CA, USA), and anti-CD140a, a marker for fibroblasts (Thermo Fisher, San Diego, CA, USA) for 30 min [35,63]. 

### 4.6. SA-β-Gal Determination by Flow Cytometry

The ß-Gal enzyme was detected in the fluorescein isothiocyanate (FITC) spectrum as described before [64]. The cells were incubated with bafilomycin A (Sigma-Aldrich, St. Louis, MO, USA) 100 nM for 1 h to induce lysosomal alkalization. The population with smaller size, complexity, and lower β-Gal activity was determined as NS. For RS and SIPS, the opposite characteristics were considered since β-Gal activity brightness was previously reported in SC [34,36].

### 4.7. CD47, MHC-I, and Calreticulin Determination by Flow Cytometry

The cell surface molecules analyzed were MHC-I (PE-Anti H-2 clone M1/42 by Biolegend, San Diego, CA, USA), Calreticulin (Anti-CalR clone PA3-900 by Thermo Fisher, San Diego, CA, USA) and its secondary PerCP-anti-rabbit IgG (ImmunoResearch Laboratories, West Grove, PA, USA), and CD47 (APC-Anti-CD47 clone miap301, Biolegend, San Diego, CA, USA). The cells were washed with PBS 1X, and then 10,000 cells were acquired on the BD Influx cytometer. The cytometry tests were performed on the same day for the NS, SR, and SIPS cells. 

The multiplex flow cytometry characteristics used to evaluate the expression of phagocytic function regulatory signals on unfixed and/or permeabilized senescent fibroblasts were the following Fluorophores: FITC, fluorescein isothiocyanate; BV510, Brillant Violet 510; PE-Cy7, phycoerythrin cyanine 7; APC, allophycocyanin; PE, phycoerythrin. The median fluorescence intensity (MdFI) of three surface molecules was compared between NS, RS, and SIPS fibroblast. 

### 4.8. Statistical Analysis

Data represent at least three independent experiments ± SEM. For the senescent parameters, an ANOVA followed by was used. The Kruskal–Wallis followed by Dunn’s multiple comparisons tests for the flow cytometry experiments were used. Data were statistically analyzed using Prism 7.0a (GraphPad Software, San Diego, CA, USA). * *p* < 0.05 was considered for statistical significance.

## Figures and Tables

**Figure 1 ijms-22-10215-f001:**
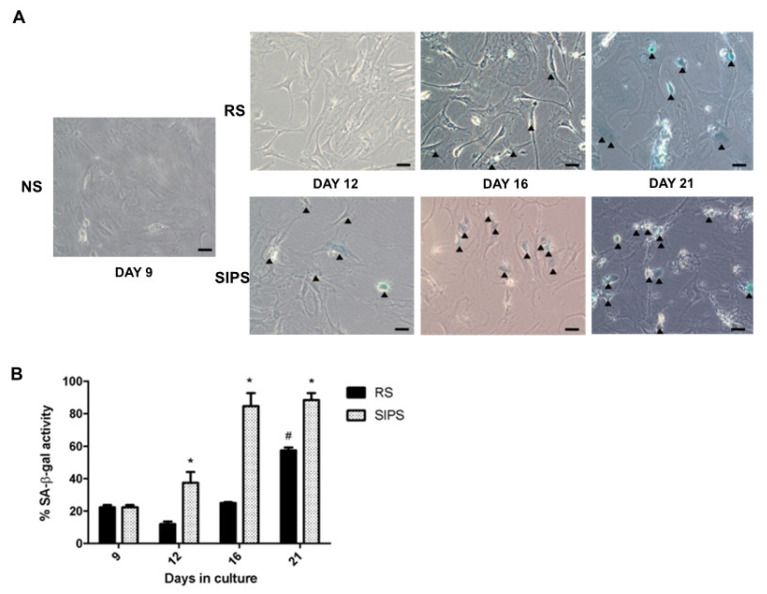
Senescence determination by SA-β-Gal assay. (**A**). Representative images of SA-β-Gal staining. NS proliferating cells, RS replicative senescence, SIPS stress-induced premature senescence. (**B**). Percentage of SA-β-Gal staining cells. The bars represent the mean ± SEM of stained cells from three independent experiments. Statistical significance # SR and * SIPS compared to NS (day 9) (*/# *p* < 0.05, ANOVA–Tukey-multiple comparisons test). Images are 40× with a scale bar indicating 20 μM. The arrows indicate positive blue staining of SC.

**Figure 2 ijms-22-10215-f002:**
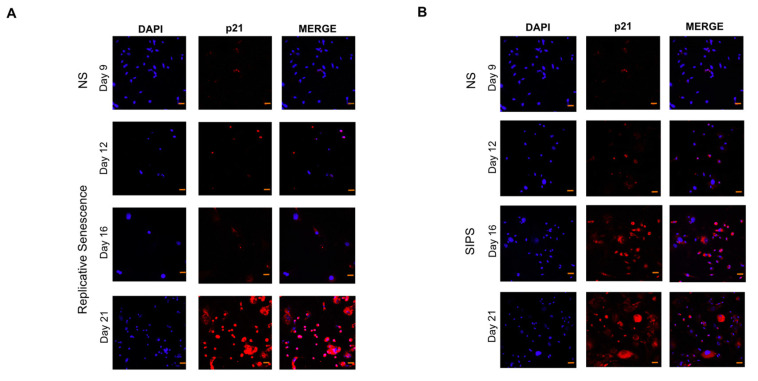
p21 immunofluorescence analysis. (**A**). The senescence marker p21 was assessed by immunofluorescence staining on days 9, 12, 16, and 21 on control fibroblasts to evaluate replicative senescence (RS): representative immunofluorescence, p21 (red) DAPI (blue). (**B**). The expression of the cell cycle inhibitor p21 was determined in H_2_O_2_-fibroblasts to evaluate stress-induced premature senescence (SIPS). Representative immunofluorescence, p21 (red) DAPI (blue). Images are 40× with a scale bar indicating 20 μM.

**Figure 3 ijms-22-10215-f003:**
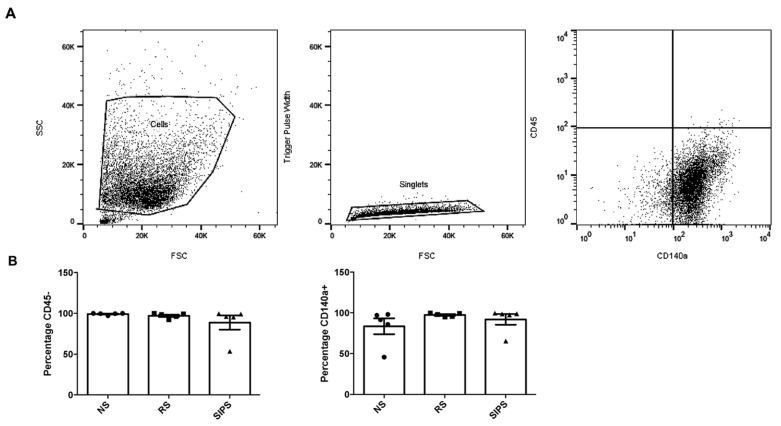
Surface immunophenotyping of fibroblasts by flow cytometry. (**A**). Gating strategy for fibroblasts of NS, RS, and SIPS. Selection of the cells on SSC vs. FSC density plot (Step 1). Single-cell selection and aggregates exclusion (Step 2). Hematopoietic cells CD45+ exclusion, and fibroblast CD140a+ selection (Step 3). (**B**). Percentage of CD45- and CD140a+ expression in NS (black circle), RS (black square), and SIPS (black triangle). No significant difference was observed between groups. Each bar represents the mean ± SEM of CD45- and CD140a+ cells. Five independent experiments were performed (Kruskal–Wallis-Dunn’s test).

**Figure 4 ijms-22-10215-f004:**
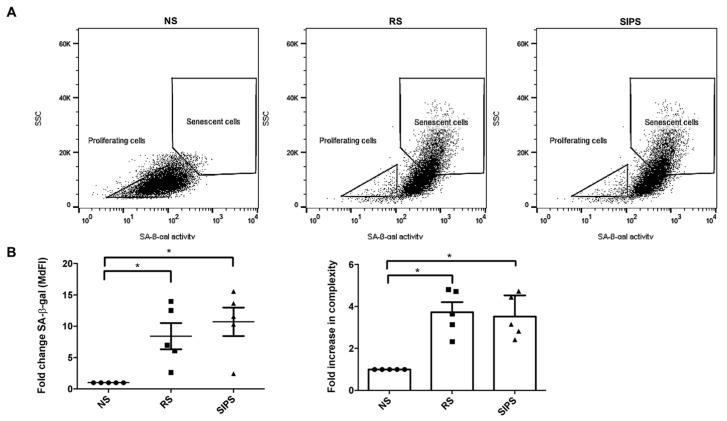
Flow cytometry gating strategy to select senescent fibroblasts. (**A**). Dot plots show that the less complex and less bright cells (low SA-ß-Gal activity) correspond to the NS or proliferating fibroblast population and are more complex and brighter (significant SA-ß-Gal activity) RS and SIPS populations. (**B**). SA-ß-Gal activity and complexity in NS (black circle), RS (black square), and SIPS (black triangle). Each bar represents the mean ± SEM of five independent experiments (* *p* < 0.05, Kruskal–Wallis-Dunn’s test).

**Figure 5 ijms-22-10215-f005:**
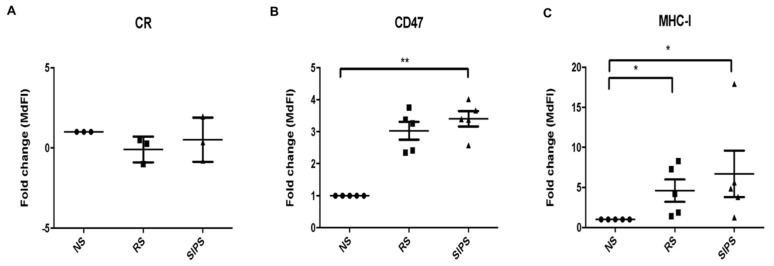
Calreticulin, CD47, and MHC-I expression. The figures represent the expression fold change of (**A**). Calreticulin, (**B**). CD47, and (**C**). MHC-I expression in NS (black circle), RS (black square), and SIPS (black triangle) performed as described in materials and methods. Each bar represents the mean ± SEM of independent experiments, three for Calreticulin and five for CD47 and MHC-I (* *p* <0.05, ** *p* < 0.01, Kruskal–Wallis-Dunn test).

**Figure 6 ijms-22-10215-f006:**
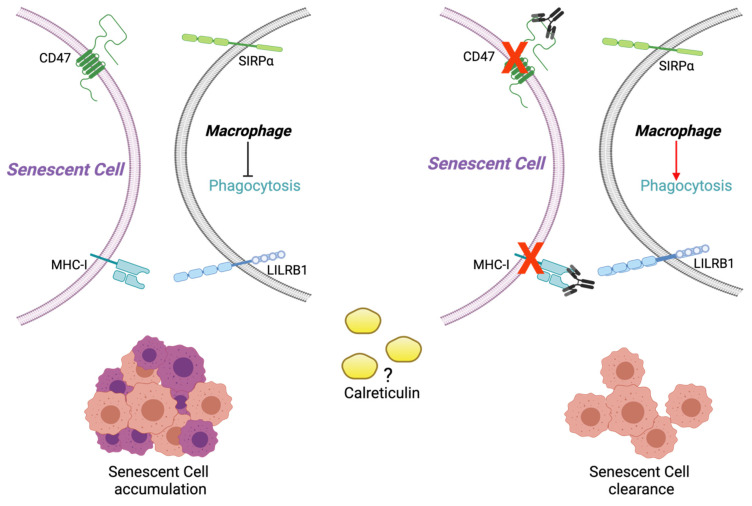
SC cells show increased CD47 and MHC class I (MHC-I) signals that mediate the “do not eat-me” antiphagocytic signals when they bind to their SIRP alpha and LILRB1 receptors, respectively. This mechanism might allow the accumulation of SC in tissues generating increased dysfunction. The use of monoclonal antibodies directed against CD47 and MHC-I could facilitate the elimination of SC. Moreover, the prophagocytic signals of Calreticulin should be further evaluated to investigate their possible participation in accumulating or eliminating SC in the tissues. Figure created with Bioender.com.

## Data Availability

Not applicable.

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
