# Peer review of "Increased CD47 and MHC Class I Inhibitory Signals Expression in Senescent CD1 Primary Mouse Lung Fibroblasts"

_ijms, 2021, doi:10.3390/ijms221910215_

Round 1
Reviewer 1 Report
To Editor:
Mercado et al examined the expression of three regulatory signalling molecules, CD47,MHC-1 and calreticulin in membrane of senescent cells. For this purpose, they used primary lung fibroblasts from two diseased senescence models, stress induced premature senescence and replicative senescence. The results show that CD47 and MHC-1 were upregulated in fibroblasts obtained from diseased models. However, no significant changes in calreticulin was observed compared to control. The authors finally conclude that CD 47 and MHC-1 can evade immune system thus avoid clearance.
The manuscript is well written, however I have concerns outlined below.
Results and Discussion:
- Line 71: more than 60%attained SIPS and RS. What do the authors thinks about the rest 40%? If incubation time is longer, can more % of cells attain senescence?
- Fig 1, Morphology images of these senescence cells could be shown? The % of SA-β-gal in SIPS is higher than RS? Can authors explain this result?
- Fig 2, it looks like expression of p21 is higher in RS model compared to SIPS?
- Line 93, it should be 2 models instead of 3.
- Line 93, the authors mention that 90% of cells were positive for CD140, but fig 3B shows almost 100% for both marker in both RS and SIPS.
- Fig 4, The authors must define complexity.
- Fig 5 and line 147, Why calreticulin did not show significant difference between control and RS/SIPS? The interpretation should be cautiously made since n=3 in RS and SIPS group. I encourage authors to increase “n” to confirm the results. Line 128, should clearly specify n=3 and n=5. Line 147, should be fig 5A
- It would be interesting to see staining of senescent cells.
- Line 170, it should be fig 5B.
- I encourage the authors to ascertain that CD47 and MHC1 are involved by using antibodies against them.
- Further, discussion about other molecules involved in senescence must be discussed.
- Methods to show senescence should be briefly described.
- Line 71: The authors should cite papers from other groups to validate the establishment of disease models of senescence.
Author Response
"Please see the attachment."

Reviewer 2 Report
Authors reported the CD47 and MHC class I inhibitory signals induction in senescent CD1 primary mouse lung fibroblasts.
- Scale bar is needed in figures.
- The meaning of */# should be explained in figure legend.
- Other senescence markers including p16, E2F, RB1 should be tested.
- Define abbreviations the first time they are mentioned in the abstract, text; also the first time they are mentioned in a table or figure.
- Keywords are not capitalized.
- There are double spaces in manuscript in line 29 "2.1 Establishment and"
- Please present the conventional senescence signal pathways in introduction section in detail.
- The figure that represents the mechanism of this study should be added.
- What is the value of this report? Please indicate the meaning of the conclusion in discussion section.
Author Response
"Please see the attachment."

Reviewer 3 Report
In this study the authors assess alterations in the expression levels of selected surface markers in senescent mouse lung fibroblasts that could serve as signals for evading immunosurveillance and clearance.
Experiments are well designed and adequately conducted and the manuscript is well-written.
For examples of some minor grammar and syntax errors, please refer to the uploaded file.
I only have two concerns:
- The authors analyzed surface markers to distinguish populations of fibroblastic cells from those of hematopoietic cells in senescent cultures. Why? It seems rational to me that characterization of fibroblasts would be performed in early-passage cells and no change would be expected in SIPS and RS fibroblasts.
- What I really miss from the markers presented to confirm senescence induction in SIPS and RS fibroblasts besides p21 over-expression and SA beta Gal positive staining is p16INK4A up-regulation. I would appreciate it if the authors included this data in the manuscript.
Author Response
"Please see the attachment."

Round 2
Reviewer 1 Report
The manuscript is significantly improved. I recommend authors to increase "n" to further strengthen the results.
Author Response
Mexico City, August 29th, 2021
Rewiever 1
IJMS
Dear Dr.,
We appreciate your comments that have improved the manuscript; all of them have already been considered in the manuscript. In respect to the last comment, this is our answer:
The experiments were evaluated in our three models: NS, RS, and SIPS. Each of these models was performed using a primary culture of lung fibroblasts from 6 individuals (neonatal mice of the CD1 strain). Each trial represents an independent experiment in flow cytometry. The statistical test that would allow us to demonstrate the significant differences for the type of data distribution and the number of groups was Kruskal-Wallis, followed by Dunn's test [1].
Five independent experiments were performed to determine CD47 and MHC-I by flow cytometry, and for calreticulin, three independent experiments were performed, and we found no changes in the statistical certainty in the determinations of the membrane surface markers evaluated.
Several articles related to the experiments described in our manuscript show that the experiments were performed in triplicate and that this number was enough to obtain statistical certainty [2-5].
- Cossarizza, A., Chang, H. D., Radbruch, A., Akdis, M., Andrä, I., Annunziato, F., Bacher, P., Barnaba, V., Battistini, L., Bauer, W. M., Baumgart, S., Becher, B., Beisker, W., Berek, C., Blanco, A., Borsellino, G., Boulais, P. E., Brinkman, R. R., Büscher, M., Busch, D. H., … Zimmermann, J. (2017). Guidelines for the use of flow cytometry and cell sorting in immunological studies. European journal of immunology, 47(10), 1584–1797. https://doi.org/10.1002/eji.201646632
- Althubiti, M., Lezina, L., Carrera, S., Jukes-Jones, R., Giblett, S. M., Antonov, A., Barlev, N., Saldanha, G. S., Pritchard, C. A., Cain, K., & Macip, S. (2014). Characterization of novel markers of senescence and their prognostic potential in cancer. Cell death & disease, 5(11), e1528. https://doi.org/10.1038/cddis.2014.489
- Madsen, S.D., Russell, K.C., Tucker, H.A. et al. (2017) Decoy TRAIL receptor CD264: a cell surface marker of cellular aging for human bone marrow-derived mesenchymal stem cells. Stem Cell Res Ther 8, 201. https://doi.org/10.1186/s13287-017-0649-4
- Biran, A., Zada, L., Abou Karam, P., Vadai, E., Roitman, L., Ovadya, Y., Porat, Z., & Krizhanovsky, V. (2017). Quantitative identification of senescent cells in aging and disease. Aging cell, 16(4), 661–671. https://doi.org/10.1111/acel.12592
- Adewoye, A.B., Tampakis, D., Follenzi, A. et al. Multiparameter flow cytometric detection and quantification of senescent cells in vitro. Biogerontology 21, 773–786 (2020). https://doi.org/10.1007/s10522-020-09893-9
I do thank you for your time, and if there is anything else I can provide in order to complete the manuscript further, please do not hesitate to contact me.
Sincerely yours,
Dra. Norma Edith López-Diazguerrero, PhD.
Departamento de Ciencias de la Salud
División de Ciencias Biológicas y de la Salud
UAM-Iztapalapa
A.P. 55-535, CDMX. 09340, México.
E-mail: norm@xanum.uam.mx
Reviewer 2 Report
Authors revised manuscript as suggestion.
However, the revisions are not applied to current version of the manuscript.
I can't fine the modified parts of cover letter in the manuscript.
I think that author didn't upload the revised version or there must be error in the system.
Please upload the new version of the manuscript.
- Scale bars in figure 1 and 2 are not applied.
- The abbreviations are not defined including MHC-I, SASP, MCP-1, MIP-1α, RANTES, etc.
- There is double space in the manuscript. For example, in line 69 "2.1 Establishment and"
- There is no new figures.
- etc....
Author Response
Rewiever 2
IJMS
Dear Dr.,
We appreciate your comments that have improved the manuscript, each one of them was correct and we answered them and included them in the revised manuscript. When we attached the revised manuscript, we checked that it was sent correctly.
In the previous reply that we sent to you, screenshots of all the changes made to the revised manuscript were added, and that was the one we sent to the Editor. We are very sorry that you could not find it, possibly there was a problem with the system. We will contact the Editor to comment on it.
Here are the changes that were made to the manuscript. I am attaching the revised manuscript in PDF:
- Scale bar is needed in figures.
Figure 1 (page 3; lines 112-113) and Figure 2 (page 4; lines 119-120).
- The meaning of */# should be explained in figure legend.
The meaning of */# was added to Figure 1(page 3; line 11-112).
- Define abbreviations the first time they are mentioned in the abstract, text; also the first time they are mentioned in a table or figure.
The abbreviations were added as the reviewer indicated.
- Keywords are not capitalized.
Keywords were capitalized: Senescent Cell Clearance Regulation; CD47; MHC Class I; Calreticulin (page 1; line 29).
- There are double spaces in manuscript in line 29 "2.1 Establishment and"
The extra spaces were eliminated (page 3; line 99).
- Please present the conventional senescence signal pathways in introduction section in detail.
Attending to the reviewer suggestion, we added a new paragraph to the introduction (pages 1-2; lines 32-51).
- The figure that represents the mechanism of this study should be added.
We performed a figure explaining our results. The figure was added to the last part of the discussion (page 9; lines 299-306).
- What is the value of this report? Please indicate the meaning of the conclusion in discussion section.
As suggested, we added a new paragraph at the beginning of the discussion (page 6, lines 168-185; page 8, lines 290-293).
I do thank you for your time, and if there is anything else I can provide in order to complete the manuscript further, please do not hesitate to contact me.
Sincerely yours,
Dra. Norma Edith López-Diazguerrero, PhD.
Departamento de Ciencias de la Salud
División de Ciencias Biológicas y de la Salud
UAM-Iztapalapa
A.P. 55-535, CDMX. 09340, México.
E-mail: norm@xanum.uam.mx
Round 3
Reviewer 1 Report
The authors have responded to all my questions and the manuscript is greatly improved.